# A Case Report and Review of the Literature of ICU Delirium

**DOI:** 10.3390/healthcare12151506

**Published:** 2024-07-30

**Authors:** Alejandro E. Brice, Roanne G. Brice

**Affiliations:** 1Department of Language, Literacy, Ed.D., Exceptional Education, and Physical Education (LLEEP) College of Education, University of South Florida, Tampa, FL 33620, USA; 2Department of Planning and Knowledge Management, College of Community Innovation and Education, University of Central Florida, Orlando, FL 32816, USA; roanne.brice@ucf.edu

**Keywords:** delirium, case reports, hallucinations, paranoid disorders

## Abstract

This case report focuses on what patients and family members may experience when a neurological trauma transpires and resultant intensive care (ICU) delirium occurs. It is the personal account of the patient (A.B.) and spouse’s (R.G.B.) perspectives when the patient (A.B) suffered a vertebral artery aneurysm and hemorrhage and experienced intensive care unit (ICU) delirium after being in the ICU for 22 days. This case report provides the patient’s and spouse’s perspectives regarding delirium, i.e., A.B.’s inability to discern reality, loss of memory, paranoia and hallucinations, agency and recovery, post-ICU syndrome, and post-traumatic stress disorder (PTSD). Clinical diagnosis by the neurosurgeon indicated delirium, with treatment consisting of sleep sedation and uninterrupted sleep. A.B. was able to regain consciousness yet experienced post-traumatic stress disorder up to one year afterward. Consistent family participation in the patient’s delirium care is crucial. Family member care and family-centered strategies are provided with implications for future research and health care.

## 1. Introduction

This case report focuses on what patients and family members may experience when a neurological trauma transpires and when resultant intensive care (ICU) delirium occurs. It is the personal account of the patient (A.B.) and spouse’s (R.G.B.) perspectives when the patient (A.B) suffered a vertebral artery aneurysm and hemorrhage and experienced ICU delirium after being in the ICU for 22 days [1,2]. Delirium is the current term for what was previously referred to as “ICU psychosis” or “ICU syndrome” [3].

## 2. Case Report

A.B. presented no health or neurological issues that are typically associated with a vertebral artery lesion [e.g., motor or sensory symptoms, dysarthria, imbalance, dizziness, tinnitus, paresthesia (tickling or prickling of the skin), homonymous hemianopsia (partial blindness), diplopia (double vision), cranial nerve palsies, or dysphagia) [4]. The only prior symptom was extreme nausea months and weeks before the aneurysmal subarachnoid hemorrhage (SAH) [5]. Subarachnoid hemorrhages are serious insults to the brain, and 30–50% percent of all SAHs result in death [6,7].

A.B. was a healthy 52-year-old male at the time of the neurological trauma (i.e., subarachnoid hemorrhage). Specifically, A.B. suffered a hemorrhage to his right vertebral artery slightly below the basilar artery and the circle of Willis (see Figure 1).

Subarachnoid hemorrhages (SAHs) are acute cerebral insults or traumas that can result from a ruptured aneurysm or traumatic brain injury [8]. Bleeding in spaces between arachnoid membranes and the pia mater of the brain is a subarachnoid hemorrhage. Consequently, blood moves into the subarachnoid space, and meningeal brain inflammation occurs [9]. SAHs also lead to blood in the cerebral spinal fluid (CSF) and resulting spinal inflammation. One to seven percent of all strokes result from subarachnoid hemorrhages [10].

A.B.’s sodium and potassium levels in the blood were extremely low for two weeks and initially required monitoring every six hours (the exact serum Na levels were never presented to the patient nor his spouse other that it was severely below the normal limit of 120 MEq/L, indicating severe hyponatremia or salt wasting). The decrease in sodium is a response to the neurological trauma and blood in the cerebrospinal fluid [11]. It is documented that long-term intensive care unit patients often endure ICU delirium [12,13,14].

As day 11 progressed, A.B. became agitated. Sleeping in the intensive care unit (ICU) is hard to do due to constant noises and interruptions for checking vital signs (i.e., blood pressure, temperature, pulse, respiratory rate, pain scale, and cognitive orientation) every few hours, along with the constant lights. Approximately 30–60% of those in the ICU experience ICU delirium [12,13]. ICU delirium is most likely due to sleep deprivation, stimulation of lights, noises, stress on the body, medications, salt wasting (i.e., hyponatremia), etc. Abraham et al. [12] stated, “Those admitted to the ICU are at greater risk for long-term disability not only because of the severity of injury, but also because of the potential lasting effects of ICU delirium (i.e., mental disorder characterized by fluctuating levels of consciousness)” (p. 2383). A.B. experienced all of the following causal risk factors for ICU delirium: (a) length of the hospital stay; (b) sleep deprivation; (c) overstimulation; (d) pain; (e) medication; (f) noise; (g) constant light; (h) lack of verbal or cognitive stimulation; (i) immobilization; (j) sensory overload; and (k) hyponatremia (salt wasting) [15]. Typically, patients who present with ten or more risk factors later present with ICU delirium [15].

## 3. Literature Review: Psychosis and Delirium

Psychosis, in general, is a collection of symptoms whereupon an individual loses touch with reality. Individuals may also exhibit delusions and/or hallucinations [16]. Julayanont and Suryadevara [17] recently stated, “The term psychosis can be described as a clinical construct where, because of severe impairment in thoughts and emotions, a person is unable to distinguish the internal experience of the mind and external reality” (p. 1682). Fuji and Ahmed [18] found psychosis to be “when an individual experiences a serious disconnect from reality. The most common symptoms associated with psychotic disorders are delusions and hallucinations” (p. 3).

An individual experiencing psychosis may experience delusions, hallucinations, and disorganized thoughts and demonstrate disorganized behaviors [17]. Psychosis can be found in cases resulting from neurodegenerative disorders. Hallucinations in psychosis can occur via any or all modalities (e.g., auditory, visual, olfactory, gustatory, tactile, kinesthetic, or somatic), while delusions are persistent and consist of false beliefs that are maintained despite contrary evidence [17]. Delusions in hospital environments would be classified as secondary delusions, i.e., those that arise from the patient’s emotional and physical state of being [18].

### 3.1. Delirium

Delirium is an acute, fluctuating, and typically reversible condition that occurs as a result of a physical illness [9]. Muralidharagopalan, Karuppasamy, and Subramanian [19] stated that delirious episodes of patients treated in ICU units without preexisting neurologic disorders or conditions are referred to as ICU delirium. Ski and O’Connell found that delirium is often misdiagnosed and inappropriately addressed in up to 94% of older patients [20].

Estimates of ICU delirium range as follows: (a) from 11% to 87% in mechanically ventilated patents and old patients [21]; (b) from 20% to 50% of patients not receiving mechanical ventilation and 60% to 80% for patients receiving mechanical ventilation [22]; and (c) from 3% to 50% of post-operative cardiac surgical older patients [23]. Hospital-based delirium results in a four-fold increase in patient mortality after two-and-a-half years post-ICU admission [12]. In addition, ICU delirium results in worse neurological outcomes at discharge and at 12 months post-ICU admission. The study done by Paixao et al. [12] found a 54% survival rate two-and-a-half years post-ICU for patients with delirium versus a 77% survival rate for post-ICU patients with no delirium. Hence, in this study, there existed a 23% difference in survival rates based on delirium for patients post-ICU. Jensen et al. [24] noted that, “Short and long term consequences of critical illness have been related to problems in the ICU and after hospital discharge [1,2,3,4]” (p. 764).

Delirium disturbances occur frequently in intensive care units (ICU) during hospital visits. Each day that the person has delirium results in a 20% increased risk of prolonged hospitalization, and “delirium is associated with a cascade of long-term cognitive impairment, functional decline, and caregiver burden” [25] (p. 1326). Delirium can be distinguished from hallucinations in that delirium and delusions are consistent and reproducible [26]. Delusions require a neuropsychological impairment leading to false beliefs and a malfunction in cognitive processes that typically would reject the false beliefs in question. Devlin et al. [27] stated, “The delirium encountered in the ICU and other settings are assumed to be equivalent pathophysiologic states. Delirium is a clinical diagnosis…” (p. 842).

Persistent cognitive impairments affect 30–80% of all ICU survivors [28]. Girard et al. [28] stated, “As many as six out of every 10 patients who survive critical illness will struggle with significant cognitive impairment months to years after their ICU stay (2)” (p. 1514). Delirium has also been associated with higher mortality rates [28]. Pisani et al. [29] stated the following:

Delirium or brain dysfunction has often been thought of as a consequence of critical illness that would resolve when the acute illness resolved. Evidence is mounting that delirium itself is a strong predictor of increased length of mechanical ventilation, longer ICU stays, increased cost, prolonged neuropsychological dysfunction, and mortality.[5,6,7,8,11,30] (p. 1095)

Intensive care unit (ICU) delirium is an acute change in mental status, difficulty with attention, disrupted perception, and/or cognitive disturbances [31]. ICU delirium subtypes consist of hypoactive, hyperactive, and mixed [21]. ICU delirium is both transient and reversible [20]. Hyperactive delirium is the most common subtype evidenced in clinical practice and is a mental state of great agitation, possible harm to self, and/or possible harm to medical personnel [22]. The hyperactive delirious patient (a) may be restless and show signs of agitation, (b) may suffer from hallucinations and delusions, and paranoia, (c) may pull at invasive lines, and/or (d) may be aggressive and/or combative [15]. Meanwhile, a patient with hypoactive delirium (a) may show slowed motor function or appear lethargic, (b) may appear confused, (c) may show poor attention span and reduced awareness, and/or (d) may be apathetic. However, hypoactive delirium is often not recognized by those in intensive care units. Patients with mixed delirium will vacillate between hyperactive and hypoactive delirium [32].

### 3.2. Causal Risk Factors for ICU Delirium

Roberts [33] stated that, “Delirium has no single cause. Rather, the development of delirium is multifactorial and is due to the brain’s non-specific reaction to disruption of the internal environment necessary for normal function” (p. 49). Numerous risk factors may hasten delirium, including predisposing factors and precipitating factors. Predisposing factors and precipitating factors include acute illness and pharmacological issues [15]. Acute illnesses include aspects such as “Intracerebral abnormalities; e.g., brain tumours/space-occupying lesions” and “Head trauma” [33] (p. 49). Aldecoa et al. [30] stated, ”There is now reasonable evidence that peripeheral inflamation and, in turn, neuroinflamation conttribute to acute deficits resembling delirium” (p. 85). Pharmacological issues arise from medications typically used for intensive care unit hospital stays (e.g., benzodiazepines) [23]. Aldecoa et al. [30] found that dexmedetomidine has shown effective anti-inflammatory and pro-autophagic responses. Sevoflurane has demonstrated both pro-inflammatory and anti-inflammatory responses, depending on the study [34]. Ghaeli et al. [34] stated that between 30% to 40% of delirium episodes can be averted. The following contributory and precipitating factors contribute to a patient’s delirium: (a) noise; (b) constant light; (c) different smells; (d) frequent interruptions; (e) sensory overload; (f) lack of verbal or cognitive stimulation; (g) social isolation; (h) sleep deprivation; and/or (i) immobilization [15,34]. Patients in the ICU on average have in excess of ten or more risk factors leading to the development of delirium.

### 3.3. Altered Mental States

Due to A.B.’s prolonged hospital stay in the intensive care unit (i.e., a total of 22 days), A.B. progressed through the following four stages of delirium over several days: (1) disorientation and loss of time and place; (b) loss of control and immobility; (c) paranoia, hallucinations, and delirium; and (d) post-traumatic stress.

Numerous coinciding factors contributed to patient disorientation, including the brain trauma, medications, and prolonged hospital stay. The subarachnoid hemorrhage caused brain inflammation and cerebral trauma, which resulted in memory, attention, and concentration issues. A.B. also received morphine, the hydromorphone Dilaudid (a narcotic opioid), and two cancer medications for nausea during this time. Due to being physically unable to move as a result of the SAH, A.B. was physically immobile for 15 days. A.B. started physical therapy 16 days post-trauma. A.B. began hallucinating, hearing voices, and becoming agitated after 11 days in the ICU unit. His hallucinations and delirium lasted for three days. Individuals with a long duration of ICU delirium (identified as those with more than one single day of ICU delirium) have worsened prognoses for mortality and long-term outcomes [35].

### 3.4. A.B.’s States of Delirium

A.B. was interviewed as a participant in a research study (a Master’s thesis at the University of Witwatersrand, South Africa) investigating ICU delirium in hospitals [36]. A.B. was designated as P1 (one of 17 participants in the study). A.B. was interviewed with his statements recorded and transcribed. Only A.B.’s transcripted personal responses were selected as the basis for this analysis. The authors separately categorized A.B.’s responses according to the following themes: (a) What is delirium; (b) Inability to discern reality; (c) Loss of memory; (d) Paranoia and hallucinations; (e) Post-ICU syndrome; and (f) Agency and recovery.

According to Roberts [33], “Delirium has no single cause. Rather, the development of delirium is multifactorial and is due to the brain’s non-specific reaction to disruption of the internal environment necessary for normal function” (p. 49). Roberts [33] also noted that optimal brain function is dependent upon functioning anatomic structures (i.e., neurophysiology), neuroelectric transmissions, and an unwavering biochemical environment. A.B.’s anatomic structures, neuroelectric transmissions, and biochemical environments had all been disrupted as a result of his subarachnoid hemorrhage. A.B. described the delirium as follows:

It started off slow and it builds up and it builds up and it builds up and it builds up and it builds up; and then there’s this tremendous explosion- crescendo, that’s overwhelming; and then I guess… it can subside, […] starting out slowly, softly, a little bit different, and then […] you have all these extraneous beats and it’s somewhat arrhythmic, and it builds up, and it’s more arrhythmia, and it builds up in a crescendo of rhythm and arrhythmia […] almost to the point of noise.(P1) [36] (p. 50)

A.B. also stated the following:

… an overwhelming bombardment of stimulation […] you’re getting overwhelmed, bombarded from the bed pumping up and down, from the noises—there’s constantly noises […]—and to being awoken every few hours: “What’s your name? When’s your birthday? Let’s check your sodium”, or “Let’s check for your blood pressure”, you know? “Let’s check this” […] and yeah, the pain […] it’s almost like the psychosis is an escape […] your brain being able to deal with the harshness of what you’re undergoing.[36] (p. 80)

### 3.5. Inability to Discern Reality

Hallucinations impair one sense of reality [37] and are defined as psychotic episodes. Hallucinations result from lack of sensory perception and can occur within any sense, such as the following: (a) visual (vision); (b) auditory (hearing); (c) tactile (touch); (d) olfactory (smell); (e) gustatory (taste); or (f) nociceptive (perception of pain), thermoceptive (sensation of temperature), proprioceptive (body position), equilibrioceptive (balance and spatial perception), kinesthetic (movement), and/or somatosensory systems (pain, hunger, and movement). A.B.’s initial hallucinations were auditory yet transcended into involving all of the above-mentioned senses. In addition, A.B. was still experiencing extreme nausea during this period of time. A.B. described the inability to discern hallucination from reality as follows:

P1 also says that “it’s almost like you toss […] all these different bits of information [into a] blender and it’s all getting mixed up and none of it makes sense but you’re in the blender”; and that “there were brief moments of reality coming into my perspective”.[36] (p. 50)

### 3.6. Loss of Memory

Time becomes distorted during periods of delirium, including memory loss [23]. Cohen et al. [23] reported on the distorted sense of time and reality during post-operative delirium among cardiac surgical patients. Amini [38] reflected on her ICU psychosis experiencing long- and short-term memory loss. Regarding attention issues, she stated, “I could not sustain attention to an activity or interaction with a person for more than a few moments. (p. 3). A.B. also said the following regarding time disorientation:

I was having hallucinations about the operation. But I had already had the operation, and I was having hallucinations about being moved into an ambulance, but I had already been moved in an ambulance […] Time was just totally out of whack.[36] (p. 89)

### 3.7. Paranoia and Hallucinations

Hallucinations and paranoia were noted by Amini [38] for the entire duration of her ICU stay. “For me, the hallucinations improved within days, but the delirium, delusions, bothersome lucid dreams, paranoia, and ensuing anxiety were intermittent yet persistent for the duration” (p. 2). Describing the hallucinatory experiences, A.B. said the below:

I wanted to be out of there […] I think I was becoming very physically agitated and trying to remove all the lines from my arms. I remember the people who were the nurses, but they were part of the hallucination […] They were officers in the air force […] And part of the hallucination was that they were trying to control me and keep me in that subjugated, uh, enforced type of state, but I always wanted to break away, I wanted to get out of there.[36] (p. 2)

A.B. described the confusion and paranoia that he experienced as follows:

I think being ill, and I think being in pain […] at that point you know something’s going on, you know you’re in a hospital; but you don’t really know. And you know a little bit about what’s going on. But […] you’re confused, and some paranoia sets in.(P1) [36] (p. 99)

A.B. also experienced Capgras syndrome, where one believes relatives to be impostors. Orum et al. [39] stated, “Capgras syndrome is a rare psychiatric disorder with colourful symptoms. The patient believes that the identities of close relatives or friends are not real but are replaced by others” (p. 110). In this instance, A.B. believed his wife to be a clone; A.B. stated, “I could tell that this was a clone, because she had a microchip in her neck” [36] (p. 102). In this hallucinatory state, A.B. recalled that “the real [spouses’ name] was supposed to rescue me” [36] (p. 107).

### 3.8. Post-ICU Syndrome and PTSD

Post-ICU syndrome is a cluster of cognitive symptoms, including anxiety, depression, and post-traumatic stress disorder. In addition, mental problems such as ICU delirium can occur. Harris [40] found that 30–50% of all patients in the ICU can suffer from post-ICU syndrome. Brice and Brice [41] found similarities to this in cases of diffuse brain injuries.

Patients in ICU are known to experience stresses and disrupted altered states, resulting in delirium. In addition, the prevalence for post-traumatic stress disorder (PTSD) rates after ICU traumas is estimated to be from 9% to 27% [42]. Physical risk factors include duration of sedation, use of benzodiazepines, and mechanical ventilation. Benzodiazepines are depressant drugs to treat insomnia, anxiety and panic disorders, depression, acute psychotic agitation, terminal agitation, seizures, neuropathic pain, skeletal muscle spasms, and alcohol withdrawal [42]. Psychological risk factors post-ICU PTSD include stress, hallucinations, delirium in ICU, and memory disturbances [43]. Wade et al. [43] identified two types of traumas that induce PTSD, i.e., traumas caused from the hallucinations and delusions (post-psychosis PTSD) and those caused from “classic PTSD” (p. 613). Classic PTSD is defined as traumas resulting from real events that occurred in the ICU, such as the patient’s injury or illness or the invasive medical procedures.

It was noted that “In the first few months after returning home, A.B. experienced post-traumatic stress disorder (PTSD). The fear of re-occurrences and also the fear of not being able to work compounded this situation” [2] (p. 7). This appears to be more of the classical type of PTSD contrasted with post-psychosis PTSD. However, A.B. also experienced stress post-ICU syndrome related to the traumas resulting from his hallucinations and delirium.

### 3.9. Agency and Recovery

Hume [36] stated the following regarding A.B.’s (or P1’s) lack of agency: “The immediate consequence of P1’s loss of agency, compounded by an absence of facts about his own circumstance, is paranoia” (p. 99). The contributing factors of illness, pain, and disorientation lead to A.B.’s lack of agency during days 11–13 post-subarachnoid hemorrhage. During days 11–13 post-subarachnoid hemorrhage, A.B. continued to have vasospasms and experiencing stresses in the ICU. He continued to be agitated. R.G.B. said, “During the several days of delirium, A.B. was attempting to pull out ICU leads, trying to get out of bed, and pulling cords out of monitoring equipment” [2] (p. 4).

### 3.10. Therapeutic Intervention

ICU delirium can be resolved using a combination of pharmacological and non-pharmacological treatments [44]. Noise and sleep deprivation are known to be high risk factors for delirium; consequently, reduction in ICU noise, uninterrupted sleep, and hydration are common non-pharmacological treatments [44]. A.B. received sleep sedation for one night yet still displayed hallucinations. The determination of delirium was clinically determined by the chief neurosurgeon. There was no clinical exam. The neurosurgeon recommended sleep anesthesia and no nightly interruptions for a complete sleep for the second night [1]. It was stated, “Once A.B. awoke, he began to make tremendous strides in returning to a more normal mental state and his health continued to improve” [1] (p. 6). In addition, A.B. began physical therapy 16 days after the SAH. A.B.’s salt wasting and the blood circulating in the cerebrospinal fluid continued to be reabsorbed into the body. A.B. experienced less severe pain, nausea, and headaches in days 16–22. A.B. was released from the hospital 22 days after the SAH.

Once A.B.’s cycle of paranoia and hallucinations were broken, his ability to discern reality and regain agency and recovery immediately increased. A.B. continued to experience memory difficulties; however, this was believed to be due more to the subarachnoid hemorrhage than the ICU delirium. Immediate resolution to the ICU delirium had been remediated; however, two months after the trauma, A.B.’s primary care physician reported that he was suffering from post-traumatic stress disorder (PTSD). A.B.’s primary complaints included panic attacks, the inability to sleep, and emotional episodes.

### 3.11. Follow-Up and Outcomes

Consistent family participation in the patient’s delirium care is crucial [45]. R.G.B.’s presence at all times was essential for A.B.’s physical state, mental state, and recuperation. Smithburger et al. [46] proclaimed that families should be active participants in delirium prevention activities. Smithburger et al. [46] specifically investigated non-pharmacological interventions. They found that “Three major themes emerged: (1) consistent family presence and participation in care, (2) improving ease of interactions between family and patient, and (3) delirium education for families” (p. e1). R.G.B. performed all of these duties. However, care for family members during the ICU delirium and post-ICU care can have deleterious effects, particularly PTSD for family member decision-makers and caregivers.

Family members and family decision-makers (FDM) of ICU patients are at high risk for PTSD following the patient’s discharge [47]. Ninety-five percent of ICU patients will need and rely on a family decision-maker (FDM) to execute major decisions, placing tremendous burden on the particular family member. In the following months post-ICU, family members are high risk of developing PTSD [47]. Lin et al. [48] reported that 75% of family members of patients with delirium suffer from consequent anxiety.

Lazarus and Folkman [49] developed the stress-appraisal-coping theory for stress management intervention. They advance two major processes in dealing with psychological stress, i.e., appraisal and coping. Appraisal relates to how the individual reacts and interprets to the stressors (e.g., harm), while coping is how the individual responds and utilizes resources to counter the situation. Flexible coping strategies are capable of adapting to the situation [50]. Petrinec et al. [51] identified the following three categories of coping behaviors: (a) problem-focused, (b) emotion-focused, and (c) avoidant-focused. Petrinec et al. [51] defined each as follows:Problem-focused coping consisted of active coping, planning, and providing instrumental support. Problem-focused coping behavior was associated with negative short-term adjustments but with positive long-term health outcomes.Emotion-focused behaviors consisted of behaviors such as receiving emotional support from others and positively changing one’s perspective, praying, and/or meditating. Emotion-focused and problem-focused behaviors were associated with less use over time as a result of improved coping with stressors.Avoidant coping behaviors consisted of behaviors such as denial, disengagement, self-blame, and/or substance abuse. Avoidance behaviors sustain and accelerate PTSD symptoms. Severe maladaptive coping behaviors are detrimental to mental health outcomes.

It appears that R.G.B. engaged in problem- and emotion-focused coping during the ICU hospital stay and immediately afterward upon both returning home. R.G.B. was focused as the only family decision-maker (FDM) and problem-solver for A.B.’s physical and psychological well-being. Emotion-focused behaviors were sustained through contact with other family members, use of a patient progress reporting website, and advocating use of self-care. R.G.B. stated, “Above all, the caregiver(s) must be aware of their physical and emotional needs and limits. Take particular care to maintain personal nourishment and needed rest” [2] (p. 80). By not engaging in avoidant behaviors, R.G.B. did not contribute to A.B.’s post-ICU delirium but minimized it, in addition to her own stress. All coping mechanisms were not needed after six months post-ICU; however, A.B. continued to make significant progress both physically and mentally after one year post-ICU. Family decision-makers need support through family-centered care (FCC).

## 4. Discussion

Family-centered care (FCC) is a partnership between family and medical staff [48] in which medical staff share information and offer support in their family member’s recovery. Allen et al. [52] stated the following:

Planning to incorporate family members or significant others into a daily rounding team may be particularly important when a high proportion of ICU patients are mechanically ventilated and receiving mind- and mood- altering agents including those targeting pain, agitation, or delirium.(p. 583)

Bohart et al. [53] identified three major themes in supporting family members as central family-centered partners in ICU delirium care. Their patient and family member interviews (*n* = 23) indicated the following themes: (1) ongoing dialogue is fundamental (communication sharing among patients, family members, and ICU staff); (2) humanizing (to be recognized as persons); and (3) equipping family to navigate (the family member’s role to advocate and support the patient). Family-centered care (FCC) may protect family members from post-ICU mental health issues [53].

Lin et al. [48] conducted a meta-analysis and systematic review regarding the efficacy of family-centered care to reduce ICU delirium. They concluded that “… FCC intervention has positive effects on reducing delirium prevalence…” (p. 1936). However, their meta-analysis study began with 3416 studies identified through database searching. After a process of screening and determining eligibility, only seven studies were included in the final analyses. This indicates a lack of studies in this area.

Davidson et al. [54] investigated 683 qualitative studies, from which 288 were analyzed further for thematic analysis. Their purpose was to update the “2007 Clinical Practice Guidelines for Support of the Family in the Patient-Centered Intensive Care Unit” (p. 4). Their report yielded 23 recommendations for family-centered care (e.g., family members participating in interdisciplinary team rounds, family members being taught how to care for critically ill members, education programs being implemented, enhanced communication with family members, or interdisciplinary family conferences to improve family trust).

Davidson et al. [54] suggest that “…clinicians use a communication approach, such as the “VALUE” mnemonic (Value family statements, Acknowledge emotions, Listen, Understand the patient as a person, Elicit Questions), during family conferences to facilitate clinician-family communication. 2C)” (p. 10). However, they report that there is lack of research tackling numerous simultaneous interventions for improving ICU care. 

In their conclusions, Davidson et al. [54] stated, “These guidelines identify the evidence base for best practices for family-centered care in the ICU. All recommendations were weak, highlighting the relative nascency of this field of research and the importance of future research to identify the most effective interventions to improve this important aspect of ICU care” (p. 4). In sum, a family-centered care (FCC) approach engages family members through the following routes: (a) dialogue and communication; (b) listening and asking questions; (c) valuing and humanizing patients and family members; (d) understanding; and (e) equipping family members [53,54].

### 4.1. Strengths and Limitations

Case reports, as with qualitative research, provide meaningful and in-depth discussions. Generally, case reports can yield high internal validity yet are extremely limited in external validity and generalization [55]. This case study presents the perspectives of the patient, spouse, and health care clinicians. A.B. and his spouse (R.G.B.) are both speech–language pathologists with advanced degrees (i.e., Ph.D.) with years of clinical experience. This unique reporting from multiple perspectives (patient, spouse, and clinician) makes this report unique. The limitations include the following: (a) one case in one situation; (b) one timeframe; (c) one hospital environment; (d) only two participant observers; and (e) other factors. The authors are aware of these limitations in that generalizability will only occur with other similar cases.

### 4.2. Patient and Spouse Perspectives

During A.B.’s hallucinatory and hyperactive delirium period, A.B. became progressively more agitated. A.B. pulled off intensive care unit leads and attempted to pull out his peripherally inserted central catheter (PICC) line. Along with the hallucinations and verbal rambling, A.B.’s spouse (R.G.B) felt that he was out of control and could potentially injure himself or others. She cited the instance where a male ICU nurse was called to restrain A.B. during these episodes. She stated, “It was terrifying to see my husband in this state” [2] (p. 6). A.B.’s spouse, R.G.B., indicated the emotional stress and exhaustion that affected her during this period.

I spoke with the neurosurgeon by phone and stressed my concerns, indicating how important it was that A.B. recover without significant disabilities. Believing that the cause was ICU delirium, the neurosurgeon prescribed sleeping medication for A.B. in hopes that the symptoms would subside. With my husband sedated and sleeping, I went home to rest after 36 h of being at my husband’s bedside.[2] (p. 6)

R.G.B. was informed that the ICU delirium, along with vasospasms within the brain, had been causing the continued issues. With this knowledge, she felt that the delirium issues would be resolved as A.B.’s health improved [41]. A.B.’s thoughts included the following: “What if these behaviors and symptoms were not completely related to ICU psychosis? What could be the cause?” [45] (column 3, line 26). Both R.G.B.’s and A.B.’s stresses resulted from A.B.’s hallucinations and delusions [43]. Patients may need to receive continued therapy and/or counseling regarding their PTSD. In addition, the advice for family members is to seek support.

Strategies that A.B. used for rehabilitation to remain cognitively active included the following: (a) word association and naming tasks (e.g., naming related words and their purpose); (b) reading low- and high-level materials as his comprehension improved (e.g., magazines to refereed journal articles); (c) writing (e.g., emails to journal articles); (d) auditory attention tasks (e.g., being able to restate main ideas); (e) divided attention tasks (e.g., focusing on specific details while ignoring erroneous information); (f) working memory and attention tasks (e.g., divided attention among multiple tasks while ignoring distractions); and (g) increasing cognitive speed (e.g., performing tasks quicker). A.B. used these strategies along with exercise (i.e., walking), appropriate sleep and rest, and a healthy diet. A.B. received spouse and family support during this time. R.G.B. facilitated A.B.’s recovery by reducing his dependence level, assisting A.B. with coping with daily and new situations, assisted with A.B.’s mood disturbances, assisted with A.B.’s fatigue and tiredness, and encouraged A.B.’s active engagement in recuperation. All of these strategies reduced both A.B.’s post-ICU syndrome. A.B. and R.G.B. continued these strategies for the next year after A.B. returned home and returned to work.

### 4.3. Conclusions

While promising, family-centered care and involvement in ICU delirium care is still understudied and needs further investigation. However, qualitative studies and case reports from patient and family perspectives are still indispensable [56]. Hartwick [56] stated, “Future research related to ICU delirium should focus on education for medical personnel regarding the conditions that lead to ICU delirium, their significance, and on effective treatment techniques” (p. 224).

After experiencing ICU delirium as a medical professional (i.e., as an occupational therapist), Amini [38] described her experiences from the patient perspective of acute care. She stated the following:

Despite my fairly extensive knowledge of our profession, coupled with 40 years of work in diverse treatment and educational settings, my eyes have been opened. I now have a new perspective and believe that living the experience of a patient in the health care context is the best way to understand what our patients experience.(p. 3)

It is anticipated that more case reports and qualitative research studies will be conducted regarding ICU delirium care. Personal case reports and qualitative research studies are indispensable and are much needed in enhancing ICU delirium education for patients, family members, and medical professionals in promoting family-centered care.

## Figures and Tables

**Figure 1 healthcare-12-01506-f001:**
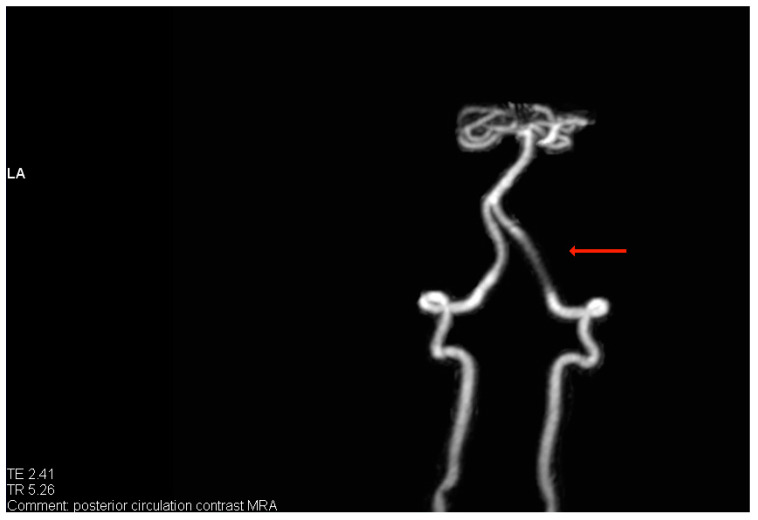
The repaired right vertebral artery (darkened area) as indicated by the red arrow. The repair consisted of placing a coil and the use of two stents. One stent was inserted inside the other. Posterior view using magnetic resonance angiogram (MRA).

## Data Availability

The original contributions presented in the study are included in the article, further inquiries can be directed to the corresponding author/s.

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
