# Peer review of "A Case Report and Review of the Literature of ICU Delirium"

_healthcare, 2024, doi:10.3390/healthcare12151506_

Round 1

Reviewer 1 Report

Comments and Suggestions for Authors

Case study is a very nice way of introducing the topic. Unfortunately, this particular case needs a lot more clinical information and insight into prognosis and all the special investigations e.g. what was the actual sodium value??? (this was thought to be a contributing factor to the cause of the delirium.

The English is very poor and needs major grammatical revision. Overall, I feel this manuscript needs extensive revision and re-submission.

Comments on the Quality of English Language

English is very poor and grammatically incorrect. Will need revision.

Author Response

  1. Reviewer 1 Comments:

    1. Comment: Case study is a very nice way of introducing the topic. Unfortunately, this particular case needs a lot more clinical information and insight into prognosis and all the special investigations e.g. what was the actual sodium value??? (this was thought to be a contributing factor to the cause of the delirium.

    Correction: This has been clarified, “serum sodium concentration levels were below 120 MEq/L indicating severe hyponatremia.”

    1. Comment: The English is very poor and needs major grammatical revision. Overall, I feel this manuscript needs extensive revision and re-submission. English is very poor and grammatically incorrect. Will need revision.

    Correction: Both authors are native English speakers. The manuscript has undergone extensive editing by both authors. In addition, the authors have used grammar checking throughout.

    3. Comment: Are the conclusions supported by the results? Can be improved.

    Correction: The authors have followed the CARE checklist (https://www.care-statement.org/checklist). Results now include 4.0 Therapeutic Intervention, 5.0 Follow-up and Outcomes, 6.0 Discussion, 7.0 Patient and Spouse Perspectives, and 8.0 Primary Lessons. The conclusions and results are supported in this case study.

    4. Comment: Does the introduction provide sufficient background and include all relevant references? Can be improved.

    Correction: Six new additional references have been added. The manuscript contains 59 references.

    ( )

    (x)

Reviewer 2 Report

Comments and Suggestions for Authors

Regarding the manuscript entitled: "A case study of ICU delirium: Patient and spouse perspectives", I would like to congratulate the authors. The report of this case helps to highlight the importance of delirium in the ICU patients and its consequences in their families. Furthermore, the authors have carried out a detailed review of the definition, risk factors, and consequences of this frequent complication of critically ill patients.

Title: I recommend indicating in the title that the current manuscript is not only a Case Report, but also a review of the literature.

Introduction: From my point of view, this section of the manuscript can be summarized. I think that the clinical case should be presented first and then the "literature review" should be carried out. Therefore, I consider that many parts of this section can be put later, in a section called "Discussion and literature review"

Patient A.B.: Instead of calling this section Patient A.B., I would call it "Case Report". From my point of view, authors should limit the information provided in this section to the information regarding the clinical case in question. It is confusing that, while the authors explain the case report, they introduce definitions and issues related to the underlying pathophysiology. From my point of view, the introduction of all this information misleads the potential reader and distances them from the chronological order of the report.

Case Report: The description of the case is adequate and detailed.

However, as this is a case report, the authors should follow the CARE guidelines, explicitly indicating each of their headings: Patient Information, Clinical Findings, Diagnostic Assessment, Therapeutic intervention, Follow-up and outcomes. It should also be added a figure or table called “Timeline”. I recommend modifying the manuscript accordingly and to add the CARE checklist (https://www.care-statement.org/checklist).

The definitions and consequences of delirium should be discussed in the "discussion" section, which I propose to be added once all the writing of the "case report" is complete.

Resolution of ICU delirium: I consider that all the information related to the case report should be stated together and discuss objective or published information in previous studies in a separate section.

Patient and Spouse Perspectives: All the information related to the case report should be stated together. Authors should discuss published information in previous studies in a separate section. The manuscript, as a case report, has become too extensive. From my point of view, this section about the “family perspectives” is crucial and I believe that the authors could focus the discussion of the case on this aspect: the perspectives of the family and the importance of involving them in the holistic management of the patient with delirium, with a view to reducing the negatives consequences that it may have.

Conclusions: Adequate.

Informed Consent Statement: Authors state that informed consent was not necessary. A.B. should consent to the publication of information regarding his illness. So, I am not sure that informed consent should be waived in a case report. I consider that the corresponding ethics committee should establish this statement.

References: I recommend including the following article: Aldecoa C, et al. Update of the European Society of Anaesthesiology and Intensive Care Medicine evidence-based and consensus-based guideline on postoperative delirium in adult patients. Eur J Anaesthesiol. 2024; 41 (2): 81 – 108. doi: 10.1097/EJA.0000000000001876. Epub 2023 Aug 30. PMID: 37599617; PMCID: PMC10763721.

Author Response

Reviewer 2 Comments:

Regarding the manuscript entitled: "A case study of ICU delirium: Patient and spouse perspectives", I would like to congratulate the authors. The report of this case helps to highlight the importance of delirium in the ICU patients and its consequences in their families. Furthermore, the authors have carried out a detailed review of the definition, risk factors, and consequences of this frequent complication of critically ill patients.

  1. Comment:

Title: I recommend indicating in the title that the current manuscript is not only a Case Report, but also a review of the literature.

Correction: The title has been changed to “A case study and review of the literature of ICU delirium: Patient and spouse perspectives.”

  1. Comment:

Introduction: From my point of view, this section of the manuscript can be summarized. I think that the clinical case should be presented first and then the "literature review" should be carried out. Therefore, I consider that many parts of this section can be put later, in a section called "Discussion and literature review"

Correction: The 2nd paragraph (page one) to page 3, has been moved. The “Case Report” has been moved before the “literature review.”

  1. Comment:

Patient A.B.: Instead of calling this section Patient A.B., I would call it "Case Report". From my point of view, authors should limit the information provided in this section to the information regarding the clinical case in question.

Correction: The subsection “2. Patient A.B.” has been changed to “2. Case Report (Patient Information).”

  1. Comment:

It is confusing that, while the authors explain the case report, they introduce definitions and issues related to the underlying pathophysiology. From my point of view, the introduction of all this information misleads the potential reader and distances them from the chronological order of the report.

Correction: The ordering has been changed as your comments. See comment #6.

Case Report: The description of the case is adequate and detailed.

  1. Comment:

However, as this is a case report, the authors should follow the CARE guidelines, explicitly indicating each of their headings: Patient Information, Clinical Findings, Diagnostic Assessment, Therapeutic intervention, Follow-up and outcomes. It should also be added a figure or table called “Timeline”. I recommend modifying the manuscript accordingly and to add the CARE checklist (https://www.care-statement.org/checklist).

Correction: The CARE checklist has been followed.

  1. Comment:

The definitions and consequences of delirium should be discussed in the "discussion" section, which I propose to be added once all the writing of the "case report" is complete.

Correction: This has been addressed with the editing changes. The authors have followed CARE checklist. Hence, the discussion includes 5.0 Follow-up and Outcomes; 6.0 Discussion, 7.0 Patient and Spouse Perspectives, and 8.0 Primary Lessons. Consequences of delirium are discussed in sections 5.0 to 8.0.

  1. Comment:

Resolution of ICU delirium: I consider that all the information related to the case report should be stated together and discuss objective or published information in previous studies in a separate section.

Correction: The CARE checklist has been followed.

  1. Comment:

Patient and Spouse Perspectives: All the information related to the case report should be stated together. Authors should discuss published information in previous studies in a separate section. The manuscript, as a case report, has become too extensive. From my point of view, this section about the “family perspectives” is crucial and I believe that the authors could focus the discussion of the case on this aspect: the perspectives of the family and the importance of involving them in the holistic management of the patient with delirium, with a view to reducing the negatives consequences that it may have.

Correction: The CARE checklist has been followed.

Conclusions: Adequate.

  1. Comment:

Informed Consent Statement: Authors state that informed consent was not necessary. A.B. should consent to the publication of information regarding his illness. So, I am not sure that informed consent should be waived in a case report. I consider that the corresponding ethics committee should establish this statement.

Correction: Informed consent was followed as per the institution’s Internal Review Board (IRB).

  1. Comment:

References: I recommend including the following article:

Aldecoa C, et al. Update of the European Society of Anaesthesiology and Intensive Care Medicine evidence-based and consensus-based guideline on postoperative delirium in adult patients. Eur J Anaesthesiol. 2024; 41 (2): 81 – 108. doi: 10.1097/EJA.0000000000001876. Epub 2023 Aug 30. PMID: 37599617; PMCID: PMC10763721.

Correction:  The above reference has been added.

Reviewer 3 Report

Comments and Suggestions for Authors

Dear authors,
Thank you very much for the opportunity to read your manuscript. A case report should adhere to the CARE checklist, on which I am going to make my comments. The patient has given informed consent?

Keywords: Must be adjusted to MeSH descriptors for correct indexing of the article, identifying diagnoses or interventions in this case report (including the keyword: "case report").

Abstract must include: What is unique about this case and what does it add to the scientific literature. The patient’s main concerns and important clinical findings. The primary diagnoses, interventions, and outcomes. What are one or more “take-away” lessons from this case report?

Figure 2 does not contribute novel information to the manuscript and can be deleted.

The discussion should inlcude: Strengths and limitations in your approach to this case.

Author Response

Dear authors,
15. Comment:

Thank you very much for the opportunity to read your manuscript. A case report should adhere to the CARE checklist, on which I am going to make my comments. The patient has given informed consent?

Correction: Consent has been given.

  1. Comment:

Keywords: Must be adjusted to MeSH descriptors for correct indexing of the article, identifying diagnoses or interventions in this case report (including the keyword: "case report").

Correction: All keywords have been changed according to MeSH descriptors.

  1. Comment:

Abstract must include: What is unique about this case and what does it add to the scientific literature. The patient’s main concerns and important clinical findings. The primary diagnoses, interventions, and outcomes. What are one or more “take-away” lessons from this case report?

Correction: The abstract has been rewritten according to the above suggestions.

  1. Comment:

Figure 2 does not contribute novel information to the manuscript and can be deleted.

Correction: Figure 2 has been deleted.

  1. Comment:

The discussion should include: Strengths and limitations in your approach to this case.

Correction: A new section in Discussions has been added i.e., 6.1 Strengths and Limitations to address this case.

Round 2

Reviewer 1 Report

Comments and Suggestions for Authors

This was a case study preceding a discussion on ICU delirium. A nice way to introduce the topic but the clinical scenario lacked extensive clinical information in terms of the clinical exam and special investigations. The laboratory values e.g. Serum Na, is not presented and therefore makes the discussion very superficial and needs much more substance.

I found the English very difficult to interpret and needs extensive editing.

I would suggest a re-submission with the appropriate changes being made.

Comments on the Quality of English Language

See above comments. 

Author Response

Reviewer One:

Does the introduction provide sufficient background and include all relevant references?

( )

(x)

( )

( )

Is the research design appropriate?

( )

(x)

( )

( )

Are the methods adequately described?

( )

(x)

( )

( )

Are the results clearly presented?

( )

(x)

( )

( )

Are the conclusions supported by the results?

( )

(x)

( )

( )

The research design is not a quantitative study. This is a qualitative case report. There no Methods section in a case report. There no Results and Conclusions section in a case report The Results and Conclusions have been reported as per the guidelines for case reports.

Comments:

This was a case study preceding a discussion on ICU delirium. A nice way to introduce the topic but the clinical scenario lacked extensive clinical information in terms of the clinical exam and special investigations.  The laboratory values e.g. Serum Na, is not presented and therefore makes the discussion very superficial and needs much more substance.

Correction:

The determination of delirium was clinically determined by the chief neurosurgeon. There was no clinical exam. This has been noted in the manuscript, lines 238-239, page 7.

The exact Serum Na levels were never presented to the patient or his spouse other that it was severely below normal limits. This has been noted in the manuscript, lines 49-50 page 2.

Comment:

I found the English very difficult to interpret and needs extensive editing.

Correction:

The manuscript has undergone rigorous re-editing. As mentioned previously, both authors are native English speakers. The reviewer does not give specific references as towhere it “needs extensive editing.” Only reviewer one is making these comments.

The first author has published 98 refereed journal articles and book chapters (67 journal articles and 31 chapters). The first author was invited to submit to this journal based on a publication in another MDPI journal. Please see below:

Brice, A., Salnaitis, C., & MacPherson, M. K. (2023). Neural activation in bilinguals and monolinguals using a word identification task. Languages, 8(3), 216. https://www.mdpi.com/2226-471X/8/3/216

Comment:

I would suggest a re-submission with the appropriate changes being made.

Correction:

Appropriate changes have been made given the feedback provided.

Reviewer Two

Does the introduction provide sufficient background and include all relevant references?

(x)

( )

( )

( )

Is the research design appropriate?

( )

(x)

( )

( )

Are the methods adequately described?

( )

(x)

( )

( )

Are the results clearly presented?

( )

(x)

( )

( )

Are the conclusions supported by the results?

(x)

( )

( )

( )

The research design is not a quantitative study. This is a qualitative case report. There no Methods section in a case report. There no Results and Conclusions section in a case report The Results and Conclusions have been reported as per the guidelines for case reports.

Comments:

I would like to congratulate the authors for the revised version of this manuscript. After modifications, manuscript’s quality has been increased.

However, authors should state 3 different sections to increase its readability:

introduction, case report, literature review and discussion. 

CARE checklist should be applicable only in the “case report” section, not throughout the whole manuscript, as authors perform an interesting “literature review”. 

Correction:

This has been addressed as the main sections are now: Introduction, Case Report, Literature Review, and Discussion.

Comment:
In my opinion, this review should focus on the “spouse perspective”, because this perspective might be the most interesting and less published up to now. 

Correction: Taking the role of the spouse as the primary perspective is a completely different case report and manuscript. 

Reviewer 2 Report

Comments and Suggestions for Authors

I would like to congratulate the authors for the revised version of this manuscript. After modifications, manuscript’s quality has been increased. However, authors should state 3 different sections to increase its readability: introduction, case report, literature review and discussion. 
CARE checklist should be applicable only in the “case report” section, not throughout the whole manuscript, as authors perform an interesting “literature review”. 
In my opinion, this review should focus on the “spouse perspective”, because this perspective might be the most interesting and less published up to now. 

Author Response

Reviewer One:

Does the introduction provide sufficient background and include all relevant references?

( )

(x)

( )

( )

Is the research design appropriate?

( )

(x)

( )

( )

Are the methods adequately described?

( )

(x)

( )

( )

Are the results clearly presented?

( )

(x)

( )

( )

Are the conclusions supported by the results?

( )

(x)

( )

( )

The research design is not a quantitative study. This is a qualitative case report. There no Methods section in a case report. There no Results and Conclusions section in a case report The Results and Conclusions have been reported as per the guidelines for case reports.

Comments:

This was a case study preceding a discussion on ICU delirium. A nice way to introduce the topic but the clinical scenario lacked extensive clinical information in terms of the clinical exam and special investigations.  The laboratory values e.g. Serum Na, is not presented and therefore makes the discussion very superficial and needs much more substance.

Correction:

The determination of delirium was clinically determined by the chief neurosurgeon. There was no clinical exam. This has been noted in the manuscript, lines 238-239, page 7.

The exact Serum Na levels were never presented to the patient or his spouse other that it was severely below normal limits. This has been noted in the manuscript, lines 49-50 page 2.

Comment:

I found the English very difficult to interpret and needs extensive editing.

Correction:

The manuscript has undergone rigorous re-editing. As mentioned previously, both authors are native English speakers. The reviewer does not give specific references as towhere it “needs extensive editing.” Only reviewer one is making these comments.

The first author has published 98 refereed journal articles and book chapters (67 journal articles and 31 chapters). The first author was invited to submit to this journal based on a publication in another MDPI journal. Please see below:

Brice, A., Salnaitis, C., & MacPherson, M. K. (2023). Neural activation in bilinguals and monolinguals using a word identification task. Languages, 8(3), 216. https://www.mdpi.com/2226-471X/8/3/216

Comment:

I would suggest a re-submission with the appropriate changes being made.

Correction:

Appropriate changes given specific feedback has been given.

Reviewer Two

Does the introduction provide sufficient background and include all relevant references?

(x)

( )

( )

( )

Is the research design appropriate?

( )

(x)

( )

( )

Are the methods adequately described?

( )

(x)

( )

( )

Are the results clearly presented?

( )

(x)

( )

( )

Are the conclusions supported by the results?

(x)

( )

( )

( )

The research design is not a quantitative study. This is a qualitative case report. There no Methods section in a case report. There no Results and Conclusions section in a case report The Results and Conclusions have been reported as per the guidelines for case reports.

Comments:

I would like to congratulate the authors for the revised version of this manuscript. After modifications, manuscript’s quality has been increased.

However, authors should state 3 different sections to increase its readability:

introduction, case report, literature review and discussion. 

CARE checklist should be applicable only in the “case report” section, not throughout the whole manuscript, as authors perform an interesting “literature review”. 

Correction:

This has been addressed as the main sections are now: Introduction, Case Report, Literature Review, and Discussion.

Comment:
In my opinion, this review should focus on the “spouse perspective”, because this perspective might be the most interesting and less published up to now. 

Correction: Taking the role of the spouse as the primary perspective is a completely different case report and manuscript.